# The Acid Ceramidase Is a SARS-CoV-2 Host Factor

**DOI:** 10.3390/cells11162532

**Published:** 2022-08-15

**Authors:** Nina Geiger, Louise Kersting, Jan Schlegel, Linda Stelz, Sofie Fähr, Viktoria Diesendorf, Valeria Roll, Marie Sostmann, Eva-Maria König, Sebastian Reinhard, Daniela Brenner, Sibylle Schneider-Schaulies, Markus Sauer, Jürgen Seibel, Jochen Bodem

**Affiliations:** 1Institute of Virology und Immunobiology, Julius-Maximilians-Universität Würzburg, Versbacher Str. 7, 97078 Würzburg, Germany; 2Institute of Organic Chemistry, Julius-Maximilians-Universität Würzburg, Am Hubland, 97074 Würzburg, Germany; 3Department of Biotechnology and Biophysics, Biocenter, Julius-Maximilians-Universität Würzburg, Am Hubland, 97074 Würzburg, Germany

**Keywords:** SARS-CoV-2, ceramides, ceramidase, fluoxetine, acid sphingomyelinase

## Abstract

SARS-CoV-2 variants such as the delta or omicron variants, with higher transmission rates, accelerated the global COVID-19 pandemic. Thus, novel therapeutic strategies need to be deployed. The inhibition of acid sphingomyelinase (ASM), interfering with viral entry by fluoxetine was reported. Here, we described the acid ceramidase as an additional target of fluoxetine. To discover these effects, we synthesized an ASM-independent fluoxetine derivative, AKS466. High-resolution SARS-CoV-2–RNA FISH and RTqPCR analyses demonstrate that AKS466 down-regulates viral gene expression. It is shown that SARS-CoV-2 deacidifies the lysosomal pH using the ORF3 protein. However, treatment with AKS488 or fluoxetine lowers the lysosomal pH. Our biochemical results show that AKS466 localizes to the endo-lysosomal replication compartments of infected cells, and demonstrate the enrichment of the viral genomic, minus-stranded RNA and mRNAs there. Both fluoxetine and AKS466 inhibit the acid ceramidase activity, cause endo-lysosomal ceramide elevation, and interfere with viral replication. Furthermore, Ceranib-2, a specific acid ceramidase inhibitor, reduces SARS-CoV-2 replication and, most importantly, the exogenous supplementation of C6-ceramide interferes with viral replication. These results support the hypotheses that the acid ceramidase is a SARS-CoV-2 host factor.

## 1. Introduction

During the last two years, a new pathogenic Coronavirus, SARS-CoV-2, became a pandemic, with high infection rates and mortality, especially in the US, South Africa, and Brazil [1]. In an unprecedented effort, several vaccines were developed and approved [2,3]. Still, the low availability of vaccines, the low vaccination rates in some developed countries, and the selection of mutants of unclear vaccine coverage endangered the success of the vaccination programs internationally [1]. Antiviral drugs offer the possibility to treat infected patients, preventing severe forms of SARS-CoV-2-associated diseases, and virus spread. Since the approval of new compounds is time-consuming, attempts to repurpose drugs already in clinical use were made. Compounds developed against the HIV-1 or Ebola viruses, such as lopinavir or remdesivir, show antiviral activity in cell culture, but fail to provide clinical benefits for SARS-CoV-2-infected patients [4,5], while other compounds, such as nafamostat, have weak effects on virus replication compared to antivirals used in HCV or HIV-1 therapy [4]. Clinical trials recently showed that the direct-acting antiviral molnupiravir reduces the risk of virus transmission, by lowering viral titers and decreasing the time of viral clearance in a clinical phase-two trial and infected mice [6,7].

We and others described that the selective serotonin reuptake receptor inhibitor (SSRI) fluoxetine blocks SARS-CoV-2 replication efficiently [8,9,10,11,12]. The relevance of this finding is strongly supported by 3D models from patients using fluoxetine and in human precision-cut lung slices (PCLS) [8]. Furthermore, a retrospective clinical study suggested that antidepressant use could be associated with a lower risk of death or intubation in patients hospitalized for COVID-19 [13]. A second, more extensive multi-center cohort study, including 3401 patients who were prescribed SSRIs, indicates that the relative risk of mortality is reduced among SARS-CoV-2 patients prescribed fluoxetine by 26% [14].

Fluoxetine inhibits the lysosomal acid sphingomyelinase (ASM) (Figure 1a), which breaks down sphingomyelin into ceramide and phosphocholine. Its activity is pH-dependent, with an optimal activity in the range of pH 4.5 to 5.0. SSRIs are protonated within the lysosome, which blocks their re-diffusion into the cytoplasm. Their local accumulation causes detachment, and thereby inactivation of ASM from the membrane of lysosomal intraluminal vesicles [15]. This also applies to acid ceramidase (AC), which catalyzes the breakdown of lysosomal ceramides (Figure 1a), released upon ASM activity, into sphingosine and fatty acids, thereby balancing ceramide levels in lysosomes, and also ensuring the subsequent export of sphingosine from this compartment after conversion into sphingosine-1-phosphate (Figure 1a) [15].

The entry pathway selection of SARS-CoV-2 depends on the protease levels displayed at the plasma membrane, such as TMPRSS2. While a high expression of TMPRSS2 protease leads to spike protein activation directly at the cell membrane, followed by rapid fusion and release at the plasma membrane, low protease levels lead to endosomal uptake [16]. The impact of sphingolipids on TMPRSS2 is unknown. Carpinteiro et al. show that ASM inhibition interferes with the expression and compartmentalization of ACE2 receptors required for the binding of SARS-CoV-2 viruses via spike protein [10]. However, the hypothesis that the ASM activity is essential for post-entry processes of the SARS-CoV-2 replication was challenged, since known SSRIs applied directly before infection, such as escitalopram, fail to inhibit SARS-CoV-2 significantly [8].

In search of mechanistic targets other than entry, we focused on the critical role of the endo-lysosomal compartment for viral replication: genome replication, late steps of viral replication, viral particle trafficking from the endoplasmic reticulum (ER), and the ER–Golgi to lysosomes, from where they are released by lysosomal exocytosis [17]. We describe a novel functionalized fluoxetine derivative, AKS466, designed not to inhibit the ASM, but efficiently inhibit SARS-CoV2 replication. Fluorescence labelling of AKS466 by click chemistry allowed us to visualize the fluoxetine derivative by high-end fluorescence microscopy in lysosomes that encapsulate viral particles. Our investigations show that treating cells with AKS466 does not prevent infection, but impairs acid ceramidase activity, followed by the entrapment of viral particles in lysosomes. Our findings identify acid ceramidase (AC) as a key enzyme in SARS-CoV-2 replication and, thus, as a promising antiviral target.

## 2. Materials and Methods

### 2.1. Chemicals and Viruses

The patient-derived SARS-CoV-2 was isolated, and the cell culture cells were previously described [8]. All reagents were purchased from commercial sources (Acros Organics (Dreieich, Germany), Alfa Aesar (Karlsruhe, Germany), BACHEM, (Bubendorf, Switzerland), Merck-Sigma Aldrich (Taufkirchen, Germany), TCI (Eschborn, Germany), VWR, (Darmstadt, Germany) and used without further purification.

### 2.2. Synthesis

Solvents were distilled prior to use. Anhydrous DMF, THF, and DCM were obtained from a solvent purification system, or purchased from Merck-Sigma Aldrich (Taufkirchen, Germany). Moisture-sensitive reactions were performed under a nitrogen atmosphere. Reactions were monitored by analytical thin-layer chromatography, using precoated silica gel plates with an ultraviolet (254 nm) indicator (ALUGRAM^®^ Xtra SIL G/UV_254_ from Macherey-Nagel, Düren, Germany). Flash column chromatography was performed with silica gel (Macherey-Nagel, Silica 60, particle size 0.040–0.063 mm). NMR spectra were measured at 295 K on a Bruker AVANCE 400 FT-NMR spectrometer. Deuterated solvents were used. The chemical shifts are reported in parts per million (ppm, δ-scale), referring to the residual solvent peak (^1^H: CDCl_3_: δ = 7.26 ppm; ^13^C: CDCl_3_: δ = 77.1 ppm). Analysis followed first order, and data are reported in the following order: chemical shift (in ppm), multiplicity (s = singlet, d = doublet, dd = doublet of doublets, t = triplet, tt = triplet of triplets, dt = doublet of triplets, q = quartet, m = multiplet, br m = broad multiplet), coupling constant (s) (in Hz), and integration. Mass spectra were recorded with a Bruker Daltonics (Billerica, MA, USA) micrOTOF-Q III spectrometer by electrospray ionization (ESI).

### 2.3. Viral RNA Isolation, Detection of Negative-Stranded Genomes and mRNAs and Viral Load Determination

Viral RNAs were isolated from 125 µL of cellular supernatants using a MagNa Pure 24 device (Roche, Mannheim, Germany), according to the manufacturer’s instructions using the pathogen 200 protocol. The RNA was eluted in 100 µL. All virus genomes were quantified with respective E-gene-specific LightMix Assays (Roche, Mannheim, Germany). Oligo d(T) (5′-TATGCGGCCGCTTTTTTTTTTTTTTTTTTTTTTTTT-3′) and an E gene sense primer (5′-ACAGGTACGTTAATAGTTAATAGCGT-3′) were used for cDNA syntheses of the E gene mRNAs and the negative sense RNAs. The amplification of these cDNAs was performed using the E gene sense and antisense Primer 5′-ACAGGTACGTTAATAGTTAATAGCGT-3′, and an E gene-specific Taqman hydrolysis probe (5″-[6FAM]ACACTAGCCATCCTTACTGCGCTTCG[BHQ1]-3″). The RNA Master Probe kit was used for amplification, as described by the manufacturer (Roche). PCR setup was performed in triplicate assays. All PCRs were performed using the LightCycler 480II device (Roche, Mannheim, Germany). Quantifications were performed with the respective cycler software.

### 2.4. Lysosome Isolation

For the lysosome isolation, 1 × 10^7^ Huh-7 or MDCK cells were treated with 30 µM AKS466, and subsequently infected with SARS-CoV-2 or Influenzavirus A. The infections were performed in duplicates. After 24 h incubation, cells were trypsinized, and the duplicates were pooled. According to the manufacturer’s instructions, lysosome isolation was performed with a lysosome isolation kit (Abcam, Cambridge, Britain). In brief, the cell pellets were resuspended in a 1.5 mL isolation buffer, and homogenized with a precooled Dounce homogenizer. After centrifugation at 500× *g* for 10 min, 375 µL supernatant was layered on a 4 mL discontinuous density gradient, followed by centrifugation for 2 h by 145,000× *g* at 4 °C. For further purification, the lysosome fraction was diluted in 3.5 mL PBS, and lysosomes were pelleted by centrifugation for 1 h by 215,000× *g* at 4 °C. The lysosome pellet was resuspended in 100 µL PBS for 15 min. Viral RNAs of 30 µL of the lysosomal fraction were isolated and quantified by RTqPCR, as described above.

### 2.5. ASM Assay

Huh-7 cells (7.5 × 10^3^) were incubated with the components of interest (AKS466, Fluoxetine) for 24 h. Afterwards, the cells were washed and lysed in 50 µL ASM-lysis buffer (250 mM Na-TC, 0.2% Na-Taucholate, 0.2% TritonX-100) by three repeated freeze-thawing cycles. The lysate was transferred to a new plate, and 10 µL of the reaction mixture was added (500 µL ASM-lysis buffer, 1.3 µL EDTA, 500 µL HMU-PC pH 5.2). The reactions were incubated for 3 h at 37 °C, and stopped with 200 µL stop-solution (0.2 M Glycine, 0.2 M NaOH, 0.25% Triton-X-100). The fluorescence was measured with the Ensight plate reader (PerkinElmer, Rodgau, Germany). To determine the substrate turnover, fluorescence was measured at an excitation of 370/404 nm/emission of 460 nm.

### 2.6. Acid Ceramidase Assay

Huh-7 cells were incubated with the compounds of interest (AKS466, fluoxetine) for 24 h. The cells were then harvested and resuspended in 40 µL ASM-lysis buffer (250 mM Na-TC, 0.2% Na-Taucholate, 0.2% TritonX-100). The cells were lysed by three repeated freeze-thawing cycles. The supernatants (10 µL) were diluted in 100 µL substrate buffer (200 mM Natrium acetate; 100 mM NaCl; 0.03% Nonidet P-40; pH 4.5) containing Cer12NBD (Cayman Chem., Ann Arbor, MICH, US), and incubated at 37 °C. The reactions were stopped after 8 h, and 24 h. Product and educts were separated on silica gel plates by chromatography. The fluorescence at 600nm was recorded for 2 min on a Li-COR Qdysse (Bad Homburg, Germany) device.

### 2.7. FISH Labelling

FISH of SARS-CoV-2-positive RNA-strand was performed, as described recently [18]. Briefly, 93 primary oligos were ordered in 96 well format (Merck-Sigma Aldrich (Taufkirchen, Germany), at a concentration of 100 µM and mixed equimolar; for oligonucleotide sequences see Appendix A. The mixture was diluted 5 times in TRIS–EDTA pH 8 (Merck-Sigma Aldrich (Taufkirchen, Germany: 93283) to obtain the final primary oligo solution, at a concentration of~20 µM. A secondary dual-Cy5-conjugated imager-oligo was ordered at a concentration of 100 µM (Merck-Sigma Aldrich (Taufkirchen, Germany),) with sequence (5′ to 3′): [Cy5]AATGCATGTCGACGAGGTCCGAGTGTAA[Cy5]. Just before labelling of SARS-CoV-2-infected cells, primary oligo solution was hybridized with the secondary imager-oligo solution in a thermocycler using the following composition and heating sequence: 2 µL primary oligo solution (20 µM) + 0.5 µL imager-oligo (100 µM) + 1 µL NEB3.1 buffer + 6.5 µL water (incubation at 85 °C for 3 min, at 65 °C for 3 min, and finally at 25 °C for 5 min). Formaldehyde-fixed and 70% ethanol permeabilized cells were washed twice with 2× SSC buffer (Sigma-Aldrich: S6639), followed by two washing steps with 2× SSC supplemented with 10% formamide. The hybridized oligos were diluted in 500 µL hybridization buffer: 100 mg/mL dextran sulfate (Sigma-Aldrich: D8906) + 10% formamide (Sigma-Aldrich: F9037) in 2× SSC buffer. Cells were labelled overnight at 37 °C in a humidified and dark chamber. Cells were washed with 2× SSC + 10% formamide, PBS, and nucleus-stained with 1 µg/mL Hoechst34580 (Merck-Sigma Aldrich Taufkirchen, Germany: 63493) for 15 min.

### 2.8. FISH Microscopy and Quantification

Imaging buffer was mixed as described previously: 2× SSC, 50 mM Tris⋅HCl pH 8, 10% (wt/vol) glucose, 2 mM Trolox (Merck-Sigma Aldrich Taufkirchen, Germany), 0.5 mg/mL glucose oxidase (Sigma-Aldrich), and 40 μg/mL catalase (Merck-Sigma Aldrich Taufkirchen, Germany) in water. To ensure stable pH values, an imaging buffer was used for a maximum of 2 h after the addition of enzymes. LatticeSIM z-stacks were acquired with Elyra7, using a C-apochromat 63×/1.2 water immersion objective and 642 nm diode laser (150 mW, 5% laser power, 50 ms exposure time, 15 phases, 196 nm slicing, z-range~10 µm) for excitation of Cy5, and Hoechst34580 was excited by the 405 nm diode laser. Detection and quantification of FISH-clusters were performed on maximum-intensity projections with an ImageJ macro (see Appendix A).

The lowest automatic threshold value used for the analysis of SARS-CoV-2-infected cells was also applied to non-infected control cells labelled by the FISH probes. Using this threshold (1100, 65,535), no single particle was detected in the maximum-intensity projections of control cells. Analysis of normal distribution, statistical significance, and graphical illustration of parameters particle area, circularity, and particle density was performed with OriginPro (OriginLab, Northampton, MA, USA). 

## 3. Results

### 3.1. Synthesis of a Fluoxetine Derivate without ASM Inhibition

We previously described the efficiency of the SSRI fluoxetine (1) to suppress SARS-CoV-2 infection in Vero, Huh-7 cells, and PCLS [8]. Based on our previous results with escitalopram, we analyzed whether ASM inhibition was essential to inhibit SARS-CoV-2 by designing and synthesizing fluoxetine derivatives, which should not inhibit this enzyme. Essentially, we planned to substitute the amino group of fluoxetine (pKa 9.5) with an amide, while keeping the aromatic core intact. This compound AKS466 (2) carries an azido group that renders the compound suitable for copper-catalyzed azide–alkyne cycloaddition (CuAAC) [19,20], and strain-promoted azide–alkyne cycloaddition (SPAAC) [21] with fluorescent dyes (click labelling) [22]. We first synthesized norfluoxetine (3) in three steps, from commercially available 3-chloro-1-phenyl-1-propanol (4) (Figure 1): we performed the first step of a Gabriel synthesis and introduced the phthalimide by nucleophilic substitution (5). We consequently introduced the CF3-aromatic ring by Mitsunobu reaction and afforded the norfluoxetine precursor (6) with a good yield. The phthalimide group is cleaved by hydrazinolysis, and produces norfluoxetine (3) in quantitative yield. Subsequently, we performed a HATU coupling with 6-azidohexanoic acid to yield the azido target molecule AKS466 (2). The compound has the same aromatic core as fluoxetine but differs in the amino side chain. In this case, the methylamine moiety is replaced by an amide. Our previous study shows that both enantiomers of fluoxetine are active against SARS-CoV-2 [8]. In analogy, we also synthesized the two enantiomers of the target molecule AKS466 starting from (S)- and (R)-3-chloro-1-phenyl-1-propanol, respectively. The synthesis followed the same four steps for each enantiomer, but it has to be considered that an inversion of the stereo-center occurs during the Mitsunobu reaction.

### 3.2. The Fluoxetine Derivative AKS466 Does Not Inhibit the ASM

First, the cytotoxicity of compound AKS466 for Huh-7 cells was tested three days after exposure. AKS466 does not exhibit toxicity at concentrations of up to 30 µM (Appendix A). We subsequently analyzed whether, as predicted, AKS466 and its R- and S-enantiomers lose the documented ability of fluoxetine to inhibit ASM. For this, Huh-7 cells were incubated with AKS466 at 15 and 30 µM concentrations for 24 h, with fluoxetine and escitalopram serving as controls. After removing the supernatants and careful washing with PBS, cells were lysed, and insoluble material was removed by centrifugation. HMU-PC was added as a substrate to the lysates for 3 h, after which its cleavage reporting ASM activity was measured with a fluorescence reader at 370/460 nm (Figure 2b). All reactions were performed in triplicates, and the experiment was repeated twice. In contrast to escitalopram and fluoxetine, AKS466 and its enantiopure stereoisomers fail to significantly suppress ASM activity (Figure 2c).

### 3.3. AKS466 Decreases Viral Replication and RNA Expression

Next, we analyzed whether AKS466 influences SARS-CoV-2 replication, in spite of it lacking ASM-inhibitory activity. Huh-7 cells were incubated with AKS466 (10 µM and 30 µM, 15 min) prior to SARS-CoV-2 infection. After three days, cell culture supernatants were collected, and viral loads were determined by RTqPCR. Strikingly, AKS466 decreases viral loads comparably to fluoxetine (Figure 3a), indicating that the derivative may act similarly efficient as an antiviral. AKS466 inhibits viral replication by more than one order of magnitude at a concentration of 30 µM in Vero cells (Appendix A). Recently, phospholipidosis-inducers, such as chloroquine, were proposed to inhibit SARS-CoV-2 such as fluoxetine [23]. However, chloroquine fails to suppress viral replication in patients and, in contrast to fluoxetine, in relevant cell lines such as Calu-3 [24] (Figure 3b). To analyze the potential inhibition by AKS466 on this cell line, we treated Calu-3 cells with AKS466, and infected the cells with 0.5 µL SARS-CoV-2. Unlike chloroquine, AKS466 decreases viral replication in Calu-3 cells, suggesting that the drug-induced phospholipidosis is not sufficient to suppress SARS-CoV-2 in a relevant cell type (Figure 3b)

Next, we investigated the impact of AKS466 on SARS-CoV-2 RNA replication and expression in cells by fluorescence RNA in situ hybridization (FISH) (Figure 3c). Therefore, Huh-7 cells were incubated with AKS466, infected with 50 µL SARS-CoV-2 for 24 h, fixed, and permeabilized. Viral RNA genomes and mRNAs are detected with complementary labelled fluorescence oligonucleotides using confocal fluorescence imaging, according to Rensen et al. [25]. SARS-CoV-2-infected cells show two specific types of (+) RNA fluorescence signals. We first detect a homogeneously distributed cytosolic background signal, most probably reflecting sub-genomic RNAs, and secondly, strongly fluorescent individual RNA particles, with sizes approaching the resolution limit of the microscope (see Appendix A). For this reason, we applied super-resolution structured illumination microscopy (SIM) to characterize the abundance and area of the small individual (+)-RNA particles at higher spatial resolution (Figure 3c). AKS466 treatment efficiently decreases the frequency of both signal intensities (Figure 3d), indicating impaired intracellular synthesis of viral RNAs. Although we do not observe differences in the number of SARS-CoV-2–RNA particles per cell, the RNA concentration is significantly reduced upon AKS466 treatment. (Figure 3d). In agreement with these observations, the size of individual (+) RNA particles is also reduced upon AKS466 treatment (Figure 3e).

To approach the mechanism of SARS-CoV-2 inhibition, we quantified the amount of viral RNAs in infected cells to directly identify the target of AKS466. Therefore, we incubated Huh-7 cells with AKS466, and infected the cells with SARS-CoV-2. After 24 h, the cells were lysed, and the cellular RNA was isolated, which was used to quantify viral RNAs by RTqPCR (Figure 4a). All infections and PCR reactions were performed in triplicates. The treatment with AKS466 significantly reduces viral RNA amounts (Figure 4a). As expected, the reduction of intracellular viral RNA is similar to the suppression of viral replication determined by quantifying viral RNA in supernatants. However, 24 h after primary infection, we would expect to see effects of reinfections. These results indicate that AKS466 inhibits SARS-CoV-2 as efficiently as fluoxetine, by blocking viral RNA synthesis or virus egress. Next, we sought to characterize the replication step at which AKS466 blocks viral replication.

### 3.4. AKS466 Enriches SARS-CoV-2 in Lysosomal Replication Compartments

SARS-CoV-2 requires lysosomes during replication and egress. It was suggested that fluoxetine inhibits the ASM in the lysosomes [10]. To analyze the localization of AKS466, we labelled the functional azido-group of AKS466 with a cell-permeable fluorescent dye by strain-promoted alkyne–azide cycloaddition (SPAAC) click-chemistry. To visualize the intracellular localization of AKS466, cells were co-stained with LysoTracker-Red and MitoTracker Red, respectively, and imaged by two-color fluorescence microscopy. Live-cell confocal fluorescence images show that similar to fluoxetine, AKS466 is localized in the lysosomes [9], suggesting that AKS466 interferes with SARS-CoV-2 replication in this compartment (Figure 4b). AKS466 is, however, not detectably enriched in the mitochondria (Figure 4b).

Since both fluoxetine and AKS466 are localized in lysosomes, we analyzed the amount of SARS-CoV-2 in this subcellular compartment. We used Huh-7 cells since they display high surface levels of the TMPRSS2 protease, which promotes SARS-CoV-2 fusion with the host plasma membrane for entry without using the endosomal pathway [16,26]. Huh-7 cells were incubated with 20 µM AKS466, and subsequently infected with SARS-CoV-2 at an MOI of 10. Cells were harvested 24 h after infection, lysed with a lysosomal lysis buffer, and lysosomes were purified by OPTI-prep gradient centrifugation. Afterwards, the lysosome-containing fractions were collected, diluted in PBS buffer (1:4), and re-pelleted. The supernatant was carefully removed, and the pellets were dissolved in 100 µL PBS buffer for 15 min on ice. Viral genomes were isolated from 30 µL of these fractions, as described above. The number of viral genomes in lysosomes from treated and untreated cells was determined by RTqPCR (Figure 4d). AKS466 treatment increases viral RNA amounts in lysosomes by about one order of magnitude, indicating that AKS466 inhibits the release of SARS-CoV-2 from this compartment and, thereby, virus egress. Interestingly, AKS466 decreases both intracellular and lysosomal viral RNA in Influenza A virus-infected MDCK cells (Appendix A), suggesting that targets for fluoxetine derivative-mediated inhibition may be virus-specific. To investigate if lysosomal fractions contain viruses on the egress route, we performed Western blotting analyses against the ER chaperon protein with an anti-GRP78 antibody, as used in a previous study [17]. Both untreated and AKS466-treated fractions contain significant amounts of the GRP78 protein (Figure 4d), indicating that the lysosomal fraction contains viruses at the egress stage. We used RTqPCR to quantify viral mRNAs and -strand genomic RNA to characterize the lysosomal fractions further. cDNAs from RNAs isolated from purified lysosomal fractions from infected Huh-7 cells were synthesized using either oligo (dT) primer to detect viral mRNAs, or an E gene-specific sense primer, to prime the minus strand and quantified by qPCR (Figure 4e,f). We found an enrichment of viral mRNA and negative stranded genome copies in the lysosomal fractions of the AKS466-treated cells, indicating that the fractions are replicative organelles, and that AKS466 treatment might impair the release of viral RNAs. The Ct-value for mRNA in the control is undetectable. In summary, treating cells with AKS466 reduces the overall amounts of viral RNA in infected cells, but accumulates viral RNAs in lysosomes.

### 3.5. AKS466 and Fluoxetine Block the Deacidification of Lysosomes

It is shown that SARS-CoV-2 deacidifies the lysosomal compartment during virus replication. This is attributed to the viral ORF3a protein, which forms an ion channel in the lysosomal membrane. Since the fluoxetine-derivative AKS466 blocks virus replication in the replicative lysosomal compartment, we sought to analyze the influence of AKS466 and fluoxetine on the pH of the lysosome. Huh-7 cells were incubated with both compounds, and the lysosomal pH was determined with live-cell imaging and pHrodo stain (Figure 5a). Both fluoxetine and AKS466 decrease the lysosomal pH significantly, indicating that the lack of viral release is correlated to a decrease in the lysosomal pH. However, the SARS ORF3a protein might overcome the effect of both compounds and deacidify the lysosomes, even in treated cells. Thus, we generated a stable ORF3a expressing Huh-7 cell line using a codon-optimized ORF3a pcDNA3.4 expression plasmid. This ORF3a cell line was treated with the compounds, and the lysosomal pH was analyzed with live-cell imaging (Figure 5b). AKS466 similarly decreases the lysosomal pH in the ORF3a cell line as in the parental Huh-7 cells, providing evidence that ORF3a could not neutralize the compound’s effect.

### 3.6. Inhibition of the Acid Ceramidase Suppresses SARS-CoV-2

Fluoxetine disrupts the ASM’s lysosomal membrane attachment, thereby inhibiting its activity and most likely that of acid ceramidase (AC). Since AKS466 is not an ASM inhibitor, we reasoned that acid ceramidase might be a prime target of both fluoxetine and AKS466, and this is important for their ability to inhibit SARS-CoV2. Huh-7 cells were treated with fluoxetine or AKS466 and incubated for 24 h. We purchased two different fluoxetine lots, since we observed lot-specific responses to SARS-CoV-2. All assays were performed in triplicates. Cells were collected and lysed in AC-lysis buffer by repeated freeze and thawing cycles. Then, the Cer_12_NBD substrate was added, and samples were incubated at 37 °C for 8 h and 24 h. Educts and products of the reactions were separated on silica gel plates and quantified by fluorescence. The experiments were repeated twice with similar results. The racemic AKS466 and its enantiopure *R*- and *S*-stereoisomers reduce AC activity by approximately 3.6-fold, while fluoxetine shows a 2-fold decrease (Figure 6a), supporting a crucial role of AC in viral replication. The influence of serotonin-reuptake inhibitors on ceramidase activity has not been reported so far. However, it is already suspected that desipramine and chlorpromazine might inhibit the AC activity, since they are shown to increase cellular ceramide amounts [15].

If AC represents a target of fluoxetine and AKS466 in SARS-CoV2 inhibition, ceranib-2 [27], a specific, small-molecule AC inhibitor, should also be effective. Notably, this substance inhibits the measles virus indirectly [28]. A total of 28 µM was determined as EC50 for AC inhibition in SKOV3 cells [27]. Thus, we decided to use 10 µM and 30 µM in SARS-CoV-2 replication assays in Huh-7 cells (these concentrations did not detectably cause cytotoxicity during the infection period; Appendix A). Cells were incubated with ceranib-2, and subsequently infected with SARS-CoV-2. The medium was removed after 24 h, and replaced by the cell culture medium containing the compound. Viral RNAs were isolated from the supernatant after 72 h, and viral genome copies were quantified by RTqPCR (Figure 6b). Ceranib-2 treatment decreases viral loads by three orders of magnitude, identifying AC as a critical enzyme in SARS-CoV-2 replication. In support of this hypothesis, marginal changes in its activity, as imposed by a low concentration of ceranib-2 (close to its EC50), effectively impacts SARS-CoV-2 replication.

### 3.7. Ceramides Inhibit SARS-CoV-2 Egress

We reasoned that one of the following mechanisms might rely on AC in SARS-CoV-2 replication. First, viral egress directly depends on AC activity; second, the virus requires one of its metabolites for replication; third, it is inhibited by ceramide accumulation. Huh-7 cell cultures were supplemented with 10 µM C6-ceramide or, for control, a non-cell permeable C6-dihydroceramide (dC6) to address the latter possibility (Figure 6b). After 2 h, the cells were infected with SARS-CoV-2 for 3 days. Then, 24 h after infection, the supernatants were replaced by a fresh medium, supplemented with 10 µM C6-/dC6-ceramide. All assays were performed in triplicates. Next, 72 h after infection, cell culture supernatants were collected, and viral RNAs were quantified by RTqPCR. The incubation with C6-ceramide substantially reduces viral RNAs in supernatants, indicating that increased C6-ceramide accumulation indeed interferes with SARS-CoV-2 replication.

To investigate changes in total cellular ceramide concentration upon treatment with AKS466, C6, and ceranib-2, we quantified the fluorescence signal of individual cells using an anti-ceramide-specific antibody (Figure 6c) [29]. Neither ceranib-2 treatment of uninfected cells nor SARS-CoV-2 infection of untreated cells influences cellular detection levels of cellular ceramide significantly. Notably, as determined by mass spectrometry, ceranib-2 does not increase ceramide content in lymphocytes [28]. However, infection of AKS466 or ceranib-2 treated cells raises ceramide levels by 1.7- and 1.4-fold, respectively. Incubation with C6 (approximately 2-fold) increases ceramide-specific signals in the cytoplasm in infected and uninfected cells. This increased ceramide detection in treated cells supports our hypothesis that ceramide concentration is critical for SARS-CoV-2 replication. Since we show that fluoxetine and AKS466 decrease the pH in the lysosomal compartment, a change in the pH should be observed if the mechanism of SARS-CoV-2 inhibition by higher ceramide concentrations is similar. Thus, we treated the ORF3a expressing Huh-7 cell line with C6 (10 µM) for 24 h and analyzed changes in the lysosomal pH with pHrodo in live-cell imaging (Figure 5b). The treatment results in a decrease in the lysosomal pH, similar to AKS466.

## 4. Discussion

In this study, we identified an additional ASM-independent inhibition of SARS-CoV-2 replication by a fluoxetine derivative compared to previous studies. Furthermore, we show that the induction of phospholipidosis is insufficient for suppressing SARS-CoV-2, since chloroquine fails to suppress the viral replication in Calu3 cells. Importantly, our study identifies AC as a novel target for therapeutic intervention against SARS-CoV-2 (Figure 7). The fluoxetine derivative AKS466 reduces viral RNA replication, similar in efficiency to fluoxetine, and causes an increase in viral RNAs in the endo-lysosomal and/or replicative compartment. Furthermore, we obtain evidence that the regulation of the ceramide concentration is critical for SARS-CoV-2.

We show that AC is an essential component in viral replication, and ceramides are a restriction factor in SARS-CoV-2 replication. Treatment with AKS466 and ceranib-2 increases ceramide detection in infected cells, along with a reduction in viral titers. In line with our observations with regard to the importance of AC in SARS-CoV-2 replication, genome-wide CRISPR screens identify the ASAH1 gene encoding AC as the host factor for SARS-CoV-2 [30]. Again, this is in line with previous results employing both genetic and pharmacological strategies. ASM-inhibition or -deficiency, respectively, are also found to elevate overall ceramide concentrations [31,32,33].

Though it argues against the role of ASM in SARS-CoV-2 lysosomal egress, our study does not preclude its importance in viral uptake shown by others previously [10]. As recently reported, SARS-CoV-2 particles are highly enriched in sphingolipids, the composition of which strongly resembles that of lipid rafts, and this proves to be essential for their infectivity [34]. Thus, ASM-catalyzed ceramide release during particle egress would be counterintuitive, and AC, in turn, may act to stabilize the lipid composition of viral particles.

Surprisingly, escitalopram does not influence SARS-CoV-2 replication, although it inhibits the ASM. Whether it acts to suppress ceramidase activity has not been analyzed so far. However, our findings that the AC is a host factor for SARS-CoV-2 offers new possible treatment strategies besides the treatment with fluoxetine by approved AC inhibitors, such as carmofur and nortriptyline. Notably, screens identified carmofur as a SARS-CoV-2 M^pro^ inhibitor with an EC50 of 24 µM [35,36]. Therefore, we predict that treatment with carmofur will inhibit both the processing of viral proteins and viral replication by AC inhibitors such as fluoxetine. Previous results with amitriptyline, with nortriptyline as the active molecule, show that it blocks virus replication in nasal tissues [37].

Our study thereby identifies AC as a host factor, critically important in SARS-CoV-2 replication, and a promising intervention target. We observed that AC inhibitors increase the ceramide content in SARS-CoV-2-infected cells, while the infection itself, except for AKS466, does not significantly influence ceramide concentrations. These data support the hypothesis that SARS-CoV-2 infection might lead to higher ceramide concentrations in infected cells, which AC counteracts.

The influence of the AC on SARS-CoV-2 replication may not be unique to Coronaviruses, as AC inhibition also impairs MV replication by as yet unknown mechanisms. These may include the depletion of sphingosine, which serves as a substrate for sphingosine kinases found to support the replication of an increasing number of viruses [38]. Notably, AC may, in turn, also act as a host restriction factor. This is shown for HSV replication in macrophages, where AC activity traps the virus in late endosomal compartments, from which efficient release is supported upon genetic ablation or the pharmacological inhibition of AC.

## Figures and Tables

**Figure 1 cells-11-02532-f001:**
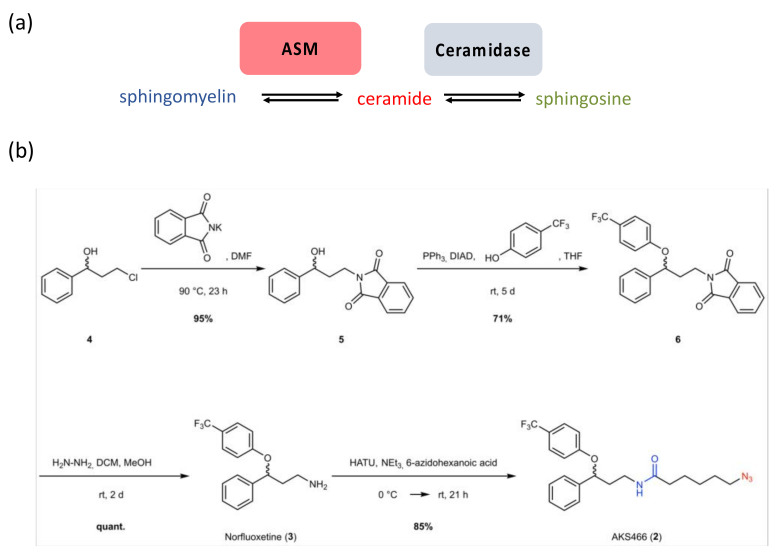
**The sphingomyelin pathway and synthesis of compound AKS466.** (**a**) Scheme sphingomyelin pathway, (**b**) introduction of an amide (blue) and a bio-orthogonal functional group (red) for click-reactions. The compound was synthesized as a racemate and as (R)- and (S)-enantiomers (see Supporting Materials). Abbreviations: DCM: dichloromethane, DIAD: diisopropyl azodicarboxylate, DMF: N,N-dimethylformamide, HATU: 1-[Bis(dimethylamino)methylene]-1H-1,2,3-triazolo [4,5-b]pyridinium 3-oxide hexafluoro-phosphate, THF: tetrahydrofurane.

**Figure 2 cells-11-02532-f002:**
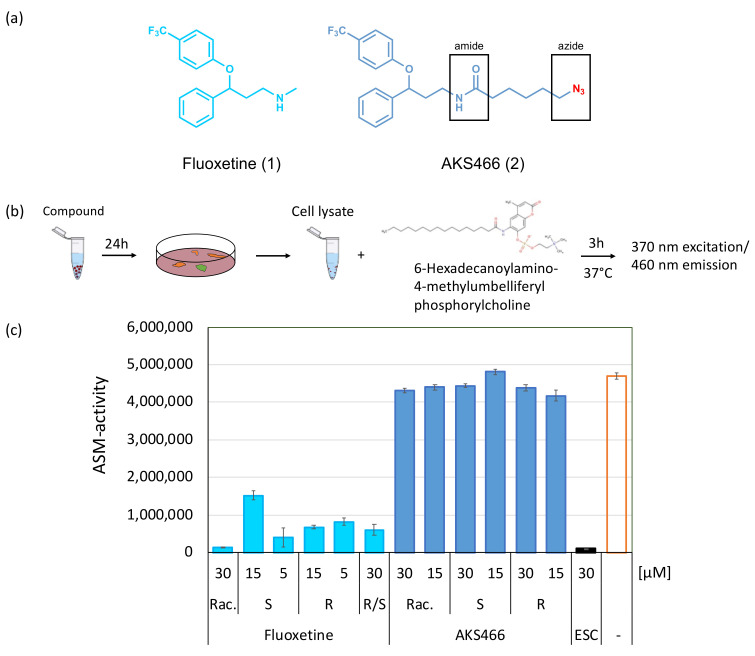
**Component AKS466 does not inhibit the ASM activity significantly**. (**a**) Molecular structures of fluoxetine and the derivative AKS466. (**b**) Scheme of the ASM assay, and (**c**) ASM assay on Huh-7 cells. The cells were incubated with the compounds for 24 h, lysed, and the HMU-PC substrate was added. The cleavage activity was determined by fluorescence. ESC: escitalopram; Rac: racemate; S and R: stereo-enantiomer; R/S: one to one R/S mixture of stereo-enantiomers.

**Figure 3 cells-11-02532-f003:**
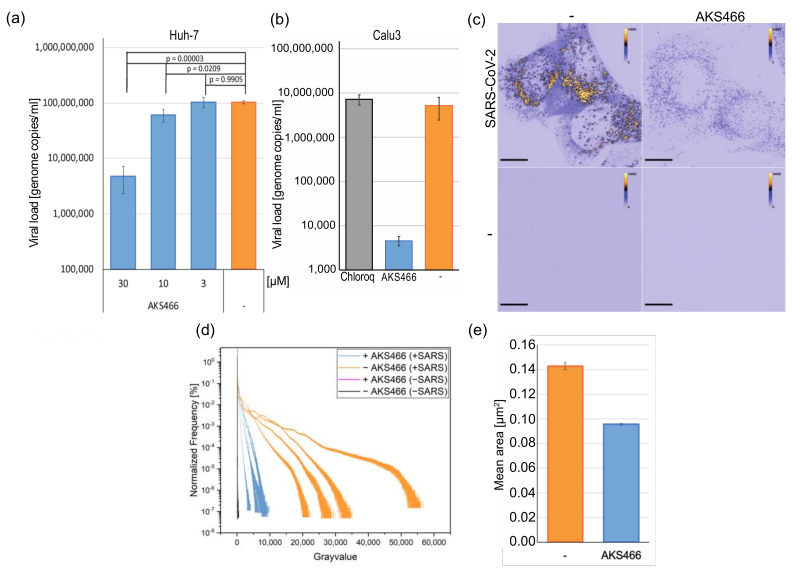
**AKS466 inhibits SARS-CoV-2 replication.** (**a**) Huh-7 were incubated with 3, 10, 30 µM AKS466, and (**b**) Calu-3 cells were incubated with 30 µM AKS466 and infected with 0.5 µL SARS-CoV-2. Cell culture supernatants were harvested after 3 days, and viral loads were determined by RTqPCR. Chloroq: Chloroquine (30 µM). (**c**) Maximum intensity projections of SARS-CoV-2 FISH confocal fluorescence images of Huh-7 cells. In infected cells (top row), fluorescence intensity decreases upon AKS466 treatment. Only a minor FISH background signal is detectable in non-infected cells (lower row). Scale bar 10 µm. (**d**) Quantification of relative fluorescence intensity frequencies of RNA FISH signal from SIM raw data from different experiments with SARS-CoV-2-infected (+SARS) and non-infected (-SARS) cells in the presence (+AKS466) and absence (-AKS466) of AKS466. (**e**) Significant reduction in the mean area of the individual (+)-RNA FISH clusters upon AKS466 treatment in SARS-CoV-2-infected cells.

**Figure 4 cells-11-02532-f004:**
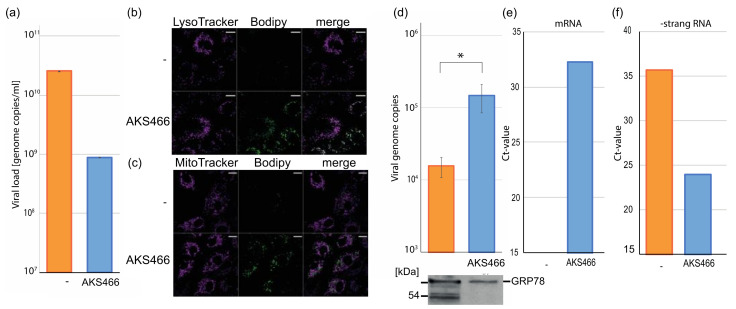
**AKS466 is localized in lysosomes and inhibits the release of SARS-CoV-2.** (**a**) AKS466 reduces the viral RNA concentration in the cells. Cells were treated and infected with AKS466, cellular RNAs were isolated, and viral transcripts were quantified by RTqPCR. (**b**,**c**) Cells were incubated with 15 µM of AKS466 for 1 h and labelled with 4 µM DBCO-BODIPY for 15 min. Lysosomes and mitochondria were co-stained with LysoTracker (**b**) or MitoTracker (**c**), respectively. The significance was calculated with the student’s *t*-test. (* *p* < 0.005) . Cells were treated with AKS466. The subcellular localization of AKS466 was visualized by linking BODIPY with a click-reaction to AKS466. Scale bar 4 µm (**d**–**f**) Huh-7 cells were treated with AKS466 and infected with SARS-CoV-2. The cells were lysed, and lysosomes were purified. (**d**) Viral RNAs, (**e**) mRNAs, and (**f**) (−) stranded genomic were quantified. Western blotting analyses for GRP78 served as the control for lysosomes on the egress pathway.

**Figure 5 cells-11-02532-f005:**
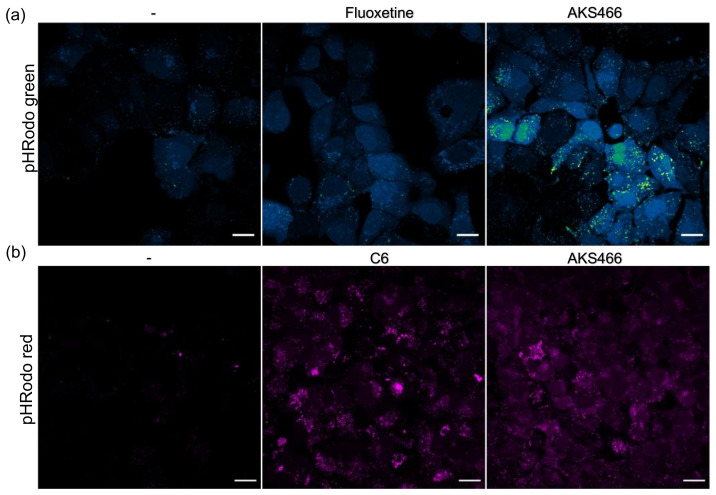
**AKS466, fluoxetine, and C6 ceramide decrease the lysosomal pH.** (**a**) Huh-7 cells were incubated with fluoxetine (5 mM) and AKS466 (20 µM). The lysosomal pH was visualized in a live-cell image with a pHrodo green stain. (**b**) The influence of ORF3a on the lysosomal pH was analyzed after AKS466 and C6 treatment of ORF3A expressing Huh-7 cells. Scale bars = 20 µm.

**Figure 6 cells-11-02532-f006:**
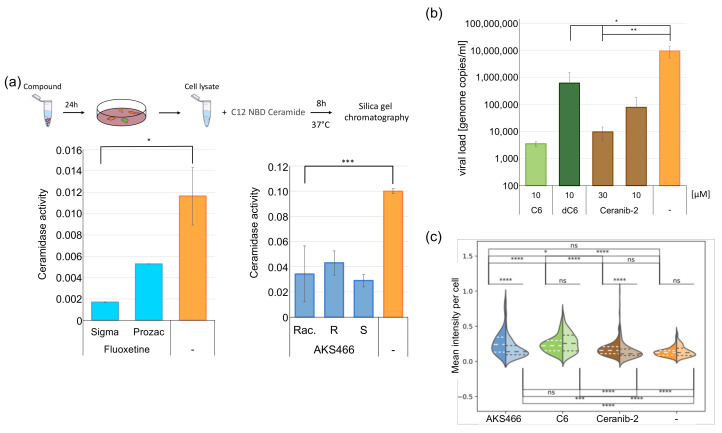
**AKS466 and fluoxetine suppress the activity of the acid ceramidase**. (**a**) Cells were incubated with the compounds for 24 h, and ceramidase activity was measured. (**b**) Ceramides inhibit SARS-CoV-2 replication. Huh-7 cells were incubated with C6 ceramides, the ceramidase inhibitor ceranib-2 or dihydroxyl C6, and infected with SARS-CoV-2. Viral loads were measured by RTqPCR. (**c**) Determination of cellular ceramides concentrations in Huh7 cells (significance: upper panel) and SARS-CoV-2-infected cells (significance: lower panel) using a ceramide-specific antibody (ns: *p* > 0.005; *: 0.005 ≥ *p* > 0.001; **: 0.001 ≥ *p* > 0.0001; ***: 0.0001 ≥ *p* > 00001; ****: 0.0001 ≥ *p*).

**Figure 7 cells-11-02532-f007:**
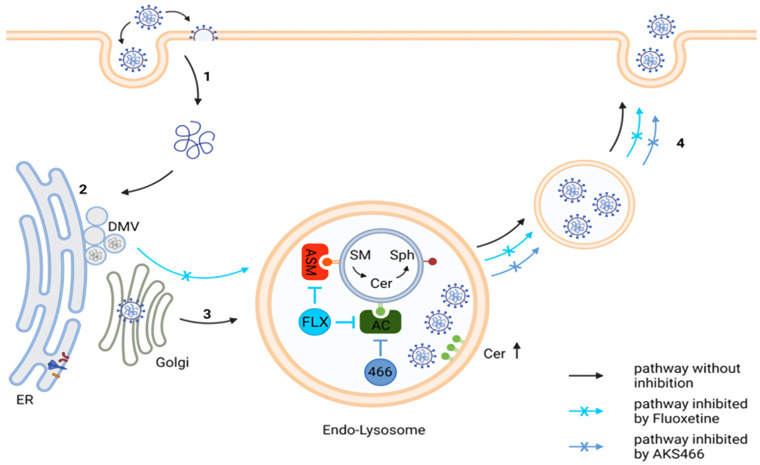
**Suggested mechanisms of fluoxetine and AKS466 inhibition**. (**1**) Viral entry by fusion pathway and release of viral RNA into the cytosol. (**2**) Viral replication and assembly. (**3**) Transport to endo-lysosomes: acid sphingomyelinase (ASM) acts at intra-luminal vesicles and degrades sphingomyelin (SM) to ceramide (Cer), which is further metabolized by acid ceramidase (AC) to sphingosine (Sph). Fluoxetine (FLX) inhibits ASM and AC, while AKS466 only inhibits AC. We suggest that inhibition of AC and ceramide accumulation leads to viral particle enrichment in endo-lysosomes and prevents release on the endo-lysosomal egress route (**4**).

## Data Availability

The lead contact will share data reported in this paper upon request.

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
