# Peer review of "The Acid Ceramidase Is a SARS-CoV-2 Host Factor"

_cells, 2022, doi:10.3390/cells11162532_

Round 1
Reviewer 1 Report
In this study the authors synthesized a derivative of fluoxetine that does not inhibit ASM and can inhibit a downstream enzyme, AC. They characterize AC as an effective antiviral enzyme against SARS2 and show that the mechanism involves trapping virus genomes in the lysosome. This is an excellent mechanistic paper and identifies a novel antiviral target against SARS2.
General comments:
It would be helpful to show the ASM-AC-lysosomal signaling/processing pathway in a figure for those who are not as familiar with it.
Can you comment on the purity of your lysosomal isolations? can other organelle/vesicle markers be detected?
Methods
- ASM and AC assays: please describe what "the components/compounds" are. please also indicate how many freeze-thaw cycles were used to lyse the cells, and how much lysis buffer and stop solution were used. It is not clear to me what the fluorescence is measuring - ASM/AC enzyme activity? - please clarify. these are interesting assays but are not commonly used and should be described so that anyone else can repeat it.
Figure 1 - I don't see a red functional group, please add! this is a nice way to highlight functional groups.
Figure 3b - how much AKS466 was used in the Calu3 cells? The inhibition is much better than in the Huh7 cells.
Figure 3, supp figure 1B, supp figure 5 etc - please indicate how much virus you used to infect the cells.
Page 9 - I would conclude that AKS466 actually does localize to the mitochondria. Although it does not overlap with mitotracker, AKS466 mimics the shape of the mitochondria. comments? perhaps AKS466 has other interesting effects.
Page 9 - it's interesting that AKS466 does not inhibit influenza A viral RNA in MDCK lysosomes... could this be repeated in a human cell line such as A549 cells? perhaps the drug simply doesn't work in canine cells....
Figure 4 e - what is the Ct value for mRNA in the control? Is it undetected?
Author Response
Reviewer 1
General comments:
It would be helpful to show the ASM-AC-lysosomal signaling/processing pathway in a figure for those who are not as familiar with it.
We added this information to Figure 1
Can you comment on the purity of your lysosomal isolations? can other organelle/vesicle markers be detected?
Probably due to the low detection limit of antibodies, we failed to detect Lamp1 and LC3 in the fractions.
Methods
- ASM and AC assays: please describe what "the components/compounds" are. please also indicate how many freeze-thaw cycles were used to lyse the cells, and how much lysis buffer and stop solution were used. It is not clear to me what the fluorescence is measuring - ASM/AC enzyme activity? - please clarify. these are interesting assays but are not commonly used and should be described so that anyone else can repeat it.
We added this information to the text.
Figure 1 - I don't see a red functional group, please add! this is a nice way to highlight functional groups.
We colourized the azido group
Figure 3b - how much AKS466 was used in the Calu3 cells? The inhibition is much better than in the Huh7 cells.
We added this information to the figure legend. We know, that the ceramidase is expressed at low levels in Huh-7 cells and assume that the expression is higher in Calu3, which might result in the better inhibition-
Figure 3, supp figure 1B, supp figure 5 etc - please indicate how much virus you used to infect the cells.
We added this information to the manuscript.
Page 9 - I would conclude that AKS466 actually does localize to the mitochondria. Although it does not overlap with mitotracker, AKS466 mimics the shape of the mitochondria. comments? perhaps AKS466 has other interesting effects.
This is an interesting observation, but since there is no overlap, we concluded that AKS466 does not enter the mitochondria.
Page 9 - it's interesting that AKS466 does not inhibit influenza A viral RNA in MDCK lysosomes... could this be repeated in a human cell line such as A549 cells? perhaps the drug simply doesn't work in canine cells....
We repeated the experiment in Calu3 cells (new panel included in the figure) since it has been shown that the Influenza virus is suppressed by Fluoxetine in this cell line. Unfortunately, we obtained the same result. However, there is a reduction in viral RNA in the cell. How the virus compensates for this remains unclear.
Figure 4 e - what is the Ct value for mRNA in the control? Is it undetected?
Yes, it is undetected. We added this information.
Reviewer 2 Report
As a physician, who works with clinical practice, and not as a laboratory scientist, I can say that the paper is well described, with laboratory details that I can't assess fully, but I can understand as being of great relevance for the evolution in the improvement of new therapeutic approaches for the SARS-CoV-2 infection.
Author Response
As a physician, who works with clinical practice, and not as a laboratory scientist, I can say that the paper is well described, with laboratory details that I can't assess fully, but I can understand as being of great relevance for the evolution in the improvement of new therapeutic approaches for the SARS-CoV-2 infection.
Thank you, for this review.
Reviewer 3 Report
This study analyzed the properties of substances that can be used to treat COVID-19.
This can be a very interesting topic.
However, since it is an analysis at the cellular level, it has potential as a therapeutic agent, but I hope that it will be developed as a therapeutic agent that can fight COVID-19 through additional analysis for clinical applications in the future.
Therefore, this paper does not appear to have any problems with publication.
Author Response
This study analyzed the properties of substances that can be used to treat COVID-19.
This can be a very interesting topic.
However, since it is an analysis at the cellular level, it has potential as a therapeutic agent, but I hope that it will be developed as a therapeutic agent that can fight COVID-19 through additional analysis for clinical applications in the future.
Therefore, this paper does not appear to have any problems with publication.
Thank you for this review
Round 2
Reviewer 1 Report
Thank you for your modifications and responses.